# Macdonald Formula, Ricci Curvature, and Concentration Locus for Classical Compact Lie Groups

Sergio Cacciatori [1,2] and Pietro Ursino [3,*]

1 Department of Science and High Technology, Università dell'Insubria, Via Valleggio 11, 22100 Como, Italy; sergio.cacciatori@uninsubria.it
2 INFN Sezione di Milano, Via Celoria 16, 20133 Milano, Italy
3 Department of Mathematics and Informatics, Università degli Studi di Catania, Viale Andrea Doria 6, 95125 Catania, Italy
* Correspondence: pietro.ursino@unict.it

**Abstract:** We consider the phenomenon of concentration of measures, which is restricted to the case of families of compact connected Lie groups. While in the literature, powerful general results regarding the existence of concentration and its relations to extremal amenability of infinite dimensional groups have been determined, there are few explicit examples, specially regarding the determination of the region where the measure concentrates. Since they can be relevant for concrete applications, both in mathematics and in physics, in the present paper, we provide a number of such examples, using compact Lie groups as basic ingredients. In particular, our strategy is to employ the Macdonald's formula, giving the volume of compact simple Lie groups, and Ricci curvature of the bi-invariant metric for analyzing a "concentration locus", which is a tool to detect where a sequence of metric, Borel measurable spaces concentrates its measure.

**Keywords:** Lie groups; invariant measures; concentration; topological dynamics

## 1. Introduction

In the last fifty years, the study of concentration of measure phenomenon has become a research field of powerful interest in different areas of mathematics. It is particularly relevant in statistic description of probabilistic phenomena, where a large number of degrees of freedom is involved and manifests itself as the "localization", under the increasing of the geometrical dimensions, of support of the measure around subregions which, strictly speaking, are zero measure subset and may find also applications in physics, see e.g., [1,2]. Intuitively, it can be understood through the following toy example: suppose we are in $\mathbb{R}^n$, looking at a ball of radius $R$. For example, we can assume it is an orange, with a very thin peel, having essentially the same specific weight of the pulp. We wonder which fraction of weight is occupied by the peel. The answer is simple, since the volume of the orange is

$$V_O = \frac{\pi^{\frac{n}{2}}}{\Gamma(n/2+1)} R^n, \tag{1}$$

while the volume of the peel is

$$V_P = 2 \frac{\pi^{\frac{n}{2}}}{\Gamma(n/2)} R^{n-1} dR. \tag{2}$$

We get that the fraction of weight occupied by the peel is

$$\varepsilon = \frac{V_P}{V_O} = n \frac{dR}{R}. \tag{3}$$

So, we see that if we fix the radius of the orange, as soon as $n$ increases, most of the mass of the orange is in a peel with thickness of the order

$$dR \approx \frac{R}{\sqrt{N}}. \tag{4}$$

In other words, all the mass (measure) appears to concentrate on the peel, despite the material being homogeneously distributed in the whole volume. If the specific weight is replaced by a uniform probability distribution, the probability tends to concentrate on the boundary of the ball, when the dimension becomes higher and higher. If in place of in $\mathbb{R}^n$ we are in a spherical world $S^n$, we would see such a uniform probability to concentrate in an equator (while we occupy one of the poles), see, e.g., [3]. Notice that in $\mathbb{R}^n$, homogeneity can be interpreted as invariance of the measure under translations. Similarly, in $S^n$, it corresponds to invariance under the action of the isometry group $SO(n+1)$. Indeed, the invariant measure can be interpreted on these spaces as induced from the invariant measure on the group itself by the action of the group on the set (for example, after fixing a point $x_0 \in S^n$, the measure of an open subset $U$ of $S^n$ can be defined by the Haar measure of the set of all $g \in SO(n+1)$ such that $gx_0 \in U$. Up to a normalization constant, this gives exactly the Lebesgue measure on $S^n$, ref. [4]). These simple considerations lead to the important problem of investigating the phenomenon of concentration of measure on spaces endowed with the action of an infinite dimensional Lie group (thought as limits of families of finite dimensional Lie groups) having an invariant normalized measure. It is clear that the relevance of such a question goes beyond statistics: if the action of the group on the space induces a measure that concentrates on a point, it is evident that such point becomes a fixed point under the action of the group. Thus, the question is strictly related to problems involving fixed point theorems under the action of infinite dimensional Lie groups [5], such as, for example, the analysis of differential equation systems. This problem has been tackled with a high level of abstraction by several authors. Starting from the pioneering research of Levy in the 1950s [3], Milman's work in the early 1970s, followed by Gromov's later work [6], the notion of Levy Family has been used to study concentration phenomenon (for an exhaustive survey on the subject, see [7–9] or the most recent [10]).

However, one of our key observations to produce new explicit examples is that this phenomenon is mostly an asymptotic effect that can be understood even in finite dimensional spaces. Looking at the concentration of measure in finite spaces, such as in the above toy example, suggests a way to understand in which way and where such concentration tends to be concretized. Indeed, in [1], we introduced the notion of Concentration Locus, which is a kind of "localized" version of concentration. In a sense, we detect in which part of the spaces, along the process of concentration, the measure concentrates.

In the present paper, we will provide explicit examples showing how the localization of invariant measures takes place for compact Lie groups. We will show different techniques apt to do it. We will make use of a formula due to Macdonald [11] for computing the volumes of compact simple Lie groups and their subgroups. The knowledge of the explicit expression for such volumes combined with the generalized Euler parametrizations of groups developed in [12] will allow us to infer concentration properties of the classical sequences of compact simple Lie groups and to calculate explicitly a concentration locus for some of them. These will be subspaces of codimension one or two, but we will also show that the concentration locus is not unique and, indeed, we are able to identify subspaces of the codimension that grow indefinitely with the dimension of the group. We also compute the Ricci curvature of such groups [13,14] and apply a Gromov–Milman's theorem [6] to deduce the Levy property for them. This method will allow us to show how to construct infinitely many concrete examples. Finally, we show how to extend our results to families of arbitrary compact connected Lie groups.

The material is organized as follows. In Section 2, we recall the background material necessary to understand the rest of the paper, including the notion of Levy Family, the no-

tion of Concentration Locus, and the Macdonald's formula. In Section 3, we show how to compute the Ricci tensor of a compact simple Lie group endowed with the natural Killing metric and then specify the results to the cases of all classical series of compact Lie groups in order to deduce the Levy property for all of them, according to Gromov–Milman's theorem. In Section 4, we study the concentration loci for all the aforementioned classical series. In particular, we make the calculations for the $SU(n)$ series very explicit, showing that it is not unique, but it can happen on subspaces of indefinitely increasing codimension. Moreover, we show that around the concentration loci, the localization of the measure is Gaussian. The analogue results for the other series are stated with the proof just sketched, being exactly a repetition of the one for $SU(n)$. At the end of the section, we show a strategy for constructing an infinite number of examples, basing once again on Gromov–Milman's theorem. In Section 5, we present our concluding remarks and perspectives.

## 2. Background and Statements

### 2.1. Levy Family and Concentration Locus

**Definition 1.** *For a set $A$ in a metric space $X$, we denote by $N^\varepsilon(A)$, $\varepsilon > 0$, its e-neighborhood. Consider a family $(X_n, \mu_n)$ with $n = 1, 2, \ldots$ of metric spaces $X_n$ with normalized borel measures $\mu_n$. We call such a family Levy if for any sequence of Borel sets $A_n \subset X_n$, $n = 1, 2, \ldots$, such that $\liminf_{n \to \infty} \mu_n(A_n) > 0$, and for every $\varepsilon > 0$, we have $\lim_{n \to \infty} \mu_n(N^\varepsilon(A)) = 1$.*

**Definition 2.** *Let $\{X_n, \mu_n\}_{n \in \mathbb{N}}$ be a family of connected metric spaces with metrics $g_n$, and $\mu_n$ be measures with respect to which open sets are measurable of non-vanishing measure. Assume the measures to be normalized, $\mu_n(X_n) = 1$. Let $\{S_n\}_{n \in \mathbb{N}}$ be a family of proper closed subsets, $S_n \subset X_n$. Fix a sequence $\{\varepsilon_n\}_{n \in \mathbb{N}}$ such that $\varepsilon_n > 0$, $\lim_{n \to \infty} \varepsilon_n = 0$, and let $\{U_n^{\varepsilon_n}\}_{n \in \mathbb{N}}$ be the sequence of tubular neighbourhoods of $S_n$ of radius $\varepsilon_n$. We say that the measure concentrates on the family $\{S_n\}$ at least at a rate of $\varepsilon_n$ if*

$$\lim_{n \to \infty} \mu_n(X_n - U_n^{\varepsilon_n}) = 0. \tag{5}$$

*We will shortly say that the measure concentrates on $S_n$ and will call it metric concentration. In particular, when $X_n$ are manifold, we call $S_n$ a concentration locus if it is contained in a submanifold of strictly positive codimension for any n. Moreover, if such a sequence $\varepsilon_n$ converges to 0 at rate k (so that $\lim_{n \to \infty} n^k \varepsilon_n = c$ for some constant c), we say that the measure concentrates on the family $\{S_n\}$ at least at rate k.*

Notice that in general, we may have $\mu_n(S_n) = 0$. Moreover, with these definitions, we do not need any notion of convergence of $S_n$ to a final subset. $S_n$ just gives a "direction of concentration". In addition, our definitions do not pretend to provide any optimality in concentration: it can happen that for a given sequence of $S_n$, there exists a sequence of proper subsets $S'_n \subset S_n$ on which we still have concentration. A special example of Definition 2 consists in the case $X_n = X$ and $S_n = S$ where $X$ is a metric space, $\{\mu_n\}_{n \in \mathbb{N}}$ is a sequence of normalized measures on $X$, compatible with the metric of $X$, and $S \subset X$ is a proper closed subset of $X$.

### 2.2. The Macdonald's Formula

Let us consider an arbitrary simple compact Lie algebra of dimension $d$ and rank $r$. It is characterized by $p = (d - r)/2$ positive roots $\alpha_i$, $i = 1, \ldots, p$ from which one can pick out a fundamental set of simple roots, say $\alpha_i$, $i = 1, \ldots, r$. To each non-vanishing root $\alpha_i$, a coroot is associated

$$\check{\alpha} = \frac{2\alpha}{(\alpha|\alpha)}, \tag{6}$$

where $(|)$ is the scalar product induced by the Killing form on the real form $H^*_{\mathbb{R}}$ of the dual $H^*$ of the Cartan subalgebra $H$. The simple coroots define a lattice, whose fundamental cell represents the fundamental torus $T^r$. The polynomial invariants of the algebra (and

groups) are generated by $r$ homogeneous polynomials of degree $d_i$, $i = 1, \ldots, r$, which are called fundamental invariants and depend on the algebra.

For such a Lie algebra, there can be several compact Lie groups. Each of them is obtained by taking the quotient of the unique compact simply connected Lie group $G$ with respect to a subgroup $\Gamma$ of the center $Z$ of $G$: $G_\Gamma = G/\Gamma$. $\Gamma$ is isomorphic to $\pi_1(G_\Gamma)$.

**Theorem 1** (Hopf). *The cohomology of a connected compact Lie group $G$ of rank $r$ over a field of characteristic 0 is that of a product of $r$ odd-dimensional spheres.*

See [15]. Indeed, such spheres have dimension $D_i = 2d_i - 1$, $i = 1, \ldots, r$, where $d_i$ represents the degrees of the fundamental invariants. The Killing form induces on a simple Lie group a unique (up to normalization) bi-invariant metric that gives to the compact group a Riemannian structure. In particular, the corresponding Riemannian volume form gives the Haar measure on the group. Normalizing the metric by fixing the length of any given simple root completely fixes, by rigidity, the entire volume of the group, which can then be computed by means of the Macdonald's formula [11,16,17]:

$$V(G_\Gamma) = \frac{1}{|\Gamma|} V(T^r) \prod_{i=1}^{r} V(S^{2d_i-1}) \prod_{i=1}^{p} (\check{\alpha}_i | \check{\alpha}_i), \tag{7}$$

where $|\Gamma|$ is the cardinality of $\Gamma$,

$$V(T^r) = |\check{\alpha}_1 \wedge \ldots \wedge \check{\alpha}_r| \tag{8}$$

and

$$V(S^{2d_i-1}) = 2 \frac{\pi^{d_i}}{(d_i - 1)!}. \tag{9}$$

## 3. Levy Property from the Ricci Tensor

We can change the property of being Levy or not, simply by rescaling the distances by $i$-dependent constants. In particular, if $X_i$, or better $(X_i, g_i)$, are compact Riemannian manifolds, and $\mu_{g_i}$ is the measure naturally associated to $g_i$, we can then consider the family

$$Y_i = (X_i, g_i, \mu_i), \qquad \mu_i = \frac{\mu_{g_i}}{\mu_{g_i}(X_i)}, \tag{10}$$

and ask whether it is Levy or not. A simple answer is given by a Corollary of the Theorem in Section 2.1 in [6]: let $\rho^{(i)}$ the Ricci tensor determined by $g_i$ and define

$$r_i = \inf \rho^{(i)}(\tau, \tau) \tag{11}$$

taken in the set of all tangent vectors of unit length. The theorem states that if

$$\lim_{i \to \infty} r_i = +\infty, \tag{12}$$

then $Y_i$ is Levy.

We will now compute the Ricci tensor for the simple groups in order to prove that the classical sequences of simple Lie groups are Levy. The Maurer–Cartan (Lie algebra valued) 1-form $\boldsymbol{j}$ over a compact Lie group $G$ is related to the bi-invariant metric $\boldsymbol{g}$ over $G$ by

$$\boldsymbol{g} = -\kappa^2 K(\boldsymbol{j} \otimes \boldsymbol{j}), \tag{13}$$

where $\kappa$ is a real normalization constant (for example, chosen so that $G$ has volume 1), and $K$ is the Killing form over $Lie(G)$, which is negative definite, since $G$ is compact. $\boldsymbol{j}$ does satisfy the Maurer–Cartan equation [18]

$$d\boldsymbol{j} + \frac{1}{2}[\boldsymbol{j},\boldsymbol{j}] = 0, \tag{14}$$

where $[,]$ is the Lie product combined with the wedge product, as usual. If we fix a basis $T_i$, $i = 1, \ldots, d$, for $\mathfrak{g} = Lie(G)$ and define the structure constants by

$$[T_i, T_j] = \sum_{k=1}^{d} c_{ij}{}^{k} T_k, \tag{15}$$

then, we can set

$$\boldsymbol{j} = \sum_{j=1}^{d} j^j T_j \tag{16}$$

and the Maurer–Cartan equation becomes

$$dj^k + \frac{1}{2} \sum_{i,j} j^i \wedge j^j c_{ij}{}^{k} = 0. \tag{17}$$

If we look at the components of $\boldsymbol{j}$ as defining a vielbein $e^i$, $i = 1, \ldots, d$, associated to a metric

$$\tilde{g}_{ij} = \delta_{ij} e^i \otimes e^j, \tag{18}$$

we see that the Maurer–Cartan equation can be seen as the structure equation for the Levi–Civita connection (in terms of the Ricci rotation coefficients):

$$de^k + \sum_j \omega^k{}_j e^j = 0, \tag{19}$$

which thus gives

$$\omega^k{}_j = \sum_i \frac{1}{2} c_{ij}{}^{k} e^i. \tag{20}$$

The curvature two form is then

$$\Omega^k{}_j = d\omega^k{}_j + \sum_l \omega^k{}_l \wedge \omega^l{}_j. \tag{21}$$

Its components $R^k{}_{jlm}$ with respect to the vielbein are thus

$$R^k{}_{jlm} = \frac{1}{4} \sum_s C_{lm}{}^{s} C_{js}{}^{k} \tag{22}$$

from which we see that the Ricci tensor has components

$$\rho_{ij} \equiv R^m{}_{imj} = -\frac{1}{4} K_{ij}, \tag{23}$$

where $K$ is the Killing form. Let us fix the compact simple Lie group $G$ and fix any basis $\{T_i\}$ for the Lie algebra in the smallest faithful representation $\sigma$. A standard choice is to assume that the basis is orthonormalized with respect to the condition (standard normalization, see Section 3.2).

$$-\frac{1}{2} Tr(\sigma(T_i) \circ \sigma(T_j)) = \delta_{ij} \tag{24}$$

which is natural since $G$ is compact. This is also a bi-invariant metric; hence, there exists a positive constant $\chi_G$ (independent from $\Gamma$) such that

$$K_{ij} = -\chi_G \delta_{ij} \tag{25}$$

so that

$$\rho_{ij} = \frac{\chi_G}{4} \delta_{ij}, \tag{26}$$

or, in coordinates,

$$\rho_{ij} = \frac{\chi_G}{4} \tilde{g}_{ij}. \tag{27}$$

The coefficients $\chi_G$ for the classical series of simple groups are computed below. We have: $\chi_{SU(n)} = n + 2$, $\chi_{SO(n)} = n - 2$ and $\chi_{USp(2n)} = 2n + 2$.

Therefore, we get the following corollary of the Gromov–Milman theorem:

**Corollary 1.** *Let*

$$Z_i = (G_i, \tilde{g}_i, \mu_i), \tag{28}$$

*where $G_i$ is any one of the classical sequences of the compact simple Lie group, which is considered in the previous section, $\tilde{g}_i$ is the corresponding standardly normalized biinvariant metric, and $\mu_i$ is the Riemannian normalized measure. Then, $\{Z_i\}_i$ is a Levy family.*

**Proof.** From (27), we get

$$r_i = \frac{\chi_G}{4}. \tag{29}$$

From the values of $\chi_G$, we get

$$r_i = \begin{cases} \frac{i+2}{4} & \text{if} \quad G = SU(i), \\ \frac{i-2}{4} & \text{if} \quad G = SO(i), \\ \frac{i+1}{2} & \text{if} \quad G = USp(2i). \end{cases} \tag{30}$$

Then, $\lim_{i \to \infty} r_i = +\infty$.  □

*3.1. Computation of $\chi_G$*

The strategy for computing the coefficient $\chi_G$ is very simple: after choosing an orthonormal basis $T_i$ in the smallest faithful representation $\sigma$, we use it to compute one of these matrices in the adjoint representation. Then

$$\chi_g = -\frac{1}{2} \text{Tr}(\text{ad}_{T_1}^2). \tag{31}$$

We will indicate with $E_{i,j}$ the elementary matrix having as the only non-vanishing element the one at line $i$ and column $j$, which is 1.

The unitary case: The representation $\sigma$ of $\mathfrak{su}(n)$ is realized by the anti-hermitian $n \times n$ matrices having a vanishing trace. A basis is given by (see [19,20]) $H_k, S_{kj}, A_{kj}$, $k = 1, \ldots, n-1, 1 \le k < j \le n$, where

$$H_k = \frac{i\sqrt{2}}{\sqrt{k^2+k}} (E_{1,1} + \ldots + E_{k,k} - kE_{k+1,k+1}), \qquad k = 1, \ldots, n-1, \tag{32}$$

$$S_{k,j} = i(E_{i,j} + E_{j,i}), \qquad k < j, \tag{33}$$

$$A_{k,j} = E_{k,j} - E_{j,k}, \qquad k < j. \tag{34}$$

Let us construct the adjoint matrix of $H_1$. The only non-vanishing commutators of $H_1$ are

$$[H_1, A_{1,2}] = 2S_{1,2}, \qquad [H_1, S_{1,2}] = -2A_{1,2}, \tag{35}$$

$$[H_1, A_{1,j}] \;\; = S_{1,j}, \qquad\qquad [H_1, S_{1,j}] = -A_{1,2}, \qquad j = 3, \dots, n. \tag{36}$$

In order to compute $(ad(H_1))^2$, we have to compute again the commutator, which gives

$$ad^2_{H_1}(A_{1,2}) \;\; = -4A_{1,2}, \qquad\qquad ad^2_{H_1}(S_{1,2}) = -4S_{1,2}, \tag{37}$$

$$ad^2_{H_1}(A_{1,j}) \;\; = -A_{1,j}, \qquad\qquad ad^2_{H_1}(S_{1,j}) = -S_{1,j}, \qquad j = 3, \dots, n. \tag{38}$$

Taking the trace, we get $\chi_{SU(n)} = n + 2$.

The orthogonal case: The representation $\sigma$ of $\mathfrak{so}(n)$ is realized by the anti-symmetric $n \times n$ matrices (see [19]). A basis is given by $A_{kj}$, $1 \le k < j \le n$, where

$$A_{k,j} \;\; = E_{i,j} - E_{j,i}, \qquad k < j. \tag{39}$$

Let us consider $ad(A_{1,2})$. The only non-vanishing commutators are

$$[A_{1,2}, A_{1,j}] = -A_{2,j}, \qquad\qquad [A_{1,2}, A_{2,j}] = A_{1,j}, \quad j = 3, \dots, n. \tag{40}$$

Iterating the commutators, we get

$$ad_{A_{1,2}}(A_{1,j}) = -A_{1,j}, \qquad\qquad ad_{A_{1,2}}(A_{2,j}) = -A_{2,j}, \quad j = 3, \dots, n. \tag{41}$$

After taking the trace, we get $\chi_{SO(n)} = n - 2$.

The symplectic case: The representation $\sigma$ of $\mathfrak{usp}(n)$ is realized by the anti-hermitian $2n \times 2n$ matrices having the form

$$\begin{pmatrix} A & B \\ C & -A^t \end{pmatrix}, \tag{42}$$

where $B$ and $C$ are symmetric (see [19]). A basis is given by

$$H_a \;\; = i(E_{a,a} - E_{a+n,a+n}), \qquad\qquad a = 1, \dots, n; \tag{43}$$

$$S^d_{ij} \;\; = \tfrac{i}{\sqrt{2}}(E_{i,j} + E_{j,i} - E_{i+n,j+n} - E_{j+n,i+n}), \qquad i < j; \tag{44}$$

$$A^d_{ij} \;\; = \tfrac{1}{\sqrt{2}}(E_{i,j} - E_{j,i} + E_{i+n,j+n} - E_{j+n,i+n}), \qquad i < j; \tag{45}$$

$$T_a \;\; = i(E_{a,a+n} + E_{a+n,a}), \qquad\qquad a = 1, \dots, n; \tag{46}$$

$$S^a_{ij} \;\; = \tfrac{i}{\sqrt{2}}(E_{i,j+n} + E_{j,i+n} + E_{i+n,j} + E_{j+n,i}), \qquad i < j; \tag{47}$$

$$U_a \;\; = (E_{a,a+n} - E_{a+n,a}), \qquad\qquad a = 1, \dots, n; \tag{48}$$

$$A^a_{ij} \;\; = \tfrac{1}{\sqrt{2}}(E_{i,j+n} + E_{j,i+n} - E_{i+n,j} - E_{j+n,i}), \qquad i < j. \tag{49}$$

We consider the adjoint representation of $H_1$. The non-vanishing commutators are

$$[H_1, S^d_{1,j}] \;\; = -A^d_{1j}, \qquad\qquad [H_1, A^d_{1,j}] = S^d_{1j}, \quad j = 2, \dots, n, \tag{50}$$

$$[H_1, T_1] \;\; = -2U_1, \qquad\qquad [H_1, U_1] = 2T_1, \tag{51}$$

$$[H_1, S^a_{1,j}] \;\; = -A^a_{1j}, \qquad\qquad [H_1, A^a_{1,j}] = S^a_{1j}, \quad j = 2, \dots, n. \tag{52}$$

Iterating the commutators, we get

$$ad^2_{H_1}(S^d_{1,j}) \;\; = -S^d_{1j}, \qquad\qquad ad^2_{H_1}(A^d_{1,j}) = -A^d_{1j}, \quad j = 2, \dots, n, \tag{53}$$

$$ad^2_{H_1}(T_1) \;\; = -4T_1, \qquad\qquad ad^2_{H_1}(U_1) = -4U_1, \tag{54}$$

$$ad^2_{H_1}(S^a_{1,j}) \;\; = -S^a_{1j}, \qquad\qquad ad^2_{H_1}(A^a_{1,j}) = -A^a_{1j}, \quad j = 2, \dots, n. \tag{55}$$

Finally, by taking the trace, we get $\chi_{USp(2n)} = 2n + 2$.

*3.2. On the Standard Normalization*

The standard normalization of the metric has a clear meaning if referred to the two-plane rotations, which are the rotations leaving fixed a codimension 2 space. These are contained in each group, and are, for example, the one generated by each of the generators $A_{k,j}$ of $SU(n)$, each of the generators of $SO(n)$, or each of the $U_a$ in the symplectic case. In order to understand its meaning, let us fix for example $A_{k,j}$ and consider the one parameter subgroup defined by

$$R \equiv R(\theta) \equiv R_{k,j}(\theta) = \exp(\theta A_{k,j}). \tag{56}$$

It represents rotations of the $k$–$j$ plane by $\theta$ and has a periodicity of $2\pi$. Let us consider the normalized metric restricted to that orbit $O \equiv O_{jk} = R([0, 2\pi])$. A simple calculation gives

$$g|_O = -\frac{1}{2}\text{Tr}(R^{-1}dR \otimes R^{-1}dR) = d\theta^2. \tag{57}$$

Thus, the total length of the whole orbit, correspondent to a continuous rotation of a round angle, is exactly $2\pi$.

## 4. Concentration Locus on Compact Lie Groups

We will start by considering the concentration of measure on compact Lie group families by direct inspection of their geometries invariant measures. Let us consider the cases of the classical series. In this case, we will prove not only that one gets Levy families, but we will also individuate at least a concentration locus.

*4.1. Concentration Locus on Simple Compact Lie Groups*

We consider the classical series of simple Lie groups. We will always mean the simply connected compact form of the groups and will consider the standard normalization for the matrices. By Corollary 1, any sequence of them is a levy family. In this section, we make a concrete calculation of a concentration locus for each of them.

4.1.1. Special Unitary Groups

The group $SU(n)$ of unitary $n \times n$ matrices with unitary determinant is a simply connected group of rank $n - 1$, and its Lie algebra is the compact form of $A_{n-1}$, that is, $\mathfrak{su}(n)$. The center is $\mathbb{Z}_n$, generated by the $n$-th roots of 1. The degrees of the fundamental invariant are $d_i = i + 1$, $i = 1, \ldots, n - 1$. The spheres generating the cohomology have dimension $D_i = 2i + 1$. With the standard normalization, a fundamental system of the simple root can be represented as follows:

One identifies isometrically $H_{\mathbb{R}}^*$ with a hyperplane of $\mathbb{R}^n$ as

$$H_{\mathbb{R}}^* \simeq \{(x_1, \ldots, x_n) \in \mathbb{R}^n | x_1 + \ldots + x_n = 0\}. \tag{58}$$

In this representation, if $\boldsymbol{e}_i$, $i = 1, \ldots, n$ is the canonical (orthonormal) basis of $\mathbb{R}^n$, and the simple roots are

$$\alpha_i = \boldsymbol{e}_i - \boldsymbol{e}_{i+1}, \qquad i = 1, \ldots, n - 1. \tag{59}$$

All roots have square length 2 and coincide with the coroots. The dimension of the group is $n^2 - 1$, so that there are $p = n(n-1)/2$ positive coroots. The volume of the torus is

$$V(T^{n-1}) = |(\boldsymbol{e}_1 - \boldsymbol{e}_2) \wedge \ldots \wedge (\boldsymbol{e}_{n-1} - \boldsymbol{e}_n)| = \sqrt{n}. \tag{60}$$

Thus, Macdonald's formula (7) gives

$$V(SU(n)) = \frac{\sqrt{n}(2\pi)^{\frac{n(n+1)}{2}-1}}{\prod_{i=1}^{n-1} i!}. \tag{61}$$

It follows that

$$\frac{V(SU(n+1))}{V(SU(n))} = \sqrt{\frac{n+1}{n}} \frac{(2\pi)^{n+1}}{n!} \sim \sqrt{\frac{2\pi}{n}} \left(\frac{2\pi e}{n}\right)^n, \tag{62}$$

so that, since $\dim SU(n+1) - \dim SU(n) = 2n+1$, we have

$$\left(\frac{V(SU(n+1))}{V(SU(n))}\right)^{\frac{1}{2n+1}} \sim \left(\frac{2\pi e}{n}\right)^{1/2}. \tag{63}$$

This is substantially the same behavior as for the spheres (of radius 1), [3], and it implies the concentration of the measure. Indeed, it means that the volume of $SU(n) \subset SU(n+1)$ grows much faster with $n$ than the volume of $SU(n+1)$. This means that if we take the normal bundle of $SU(n)$ in $SU(n+1)$ and take a neighborhood $\mathcal{T}_n$ of $SU(n)$ of radius $\varepsilon$ in the normal directions, we get for the volume of this neighborhood

$$\frac{V(SU(n+1))}{\mathcal{T}_n} \sim \sqrt{\frac{2\pi}{n}} \left(\frac{2\pi e}{n\varepsilon^2}\right)^n \frac{1}{\varepsilon}, \tag{64}$$

which for any given $\varepsilon$ decreases to 0 when $n \to \infty$. However, it does not give us direct information on how the concentration sets move. A more precise result is the following.

**Proposition 1.** *Consider the family of simple Lie groups $SU(n+1)$ endowed with the usual biinvariant metric. Let us consider the Hopf structure of $SU(n+1)$, t.i. $U(n) \hookrightarrow SU(n+1) \longrightarrow \mathbb{CP}^n$. Let $S_n$ be the hyperplane at infinity in $\mathbb{CP}^n$, and*

$$\iota : S_n \hookrightarrow \mathbb{CP}^n \tag{65}$$

*the corresponding embedding. Finally, let $\mu_n$ be the normalized invariant measure on $SU(n+1)$. Then, after looking at $SU(n+1)$ as a $U(n)$-fibration over $\mathbb{CP}^n$, the invariant measure concentrates on the real codimension 2 subvariety*

$$\Sigma_n = \iota^*(SU(n+1)), \tag{66}$$

*in the sense of Definition 2, with constant $\varepsilon$.*

**Proof.** Recall that $U(n) \subset SU(n+1)$ is a maximal proper Lie subgroup and $\mathbb{CP}^n = SU(n+1)/U(n)$ (and $SU(n) \subset U(n)$). Therefore, one expects for the measure $\mu_{SU(n+1)}$ to factorize as

$$d\mu_n = d\mu_{\mathbb{CP}^n} \times d\mu_{U(n)}. \tag{67}$$

Now, $\mathbb{CP}^n \simeq S^{2n+1}/U(1)$ and the natural metric over it is the Fubini–Study metric that is invariant under the action of the whole $SU(n+1)$ group. Thus, we expect the measure $d\mu_{\mathbb{CP}^n}$, inherited from the whole invariant measure, to be the Riemannian volume form corresponding to the Fubini–Study metric. On the other hand, the relation between $\mathbb{CP}^n$ and $S^{2n+1}$ suggests that the concentration of the measure of $d\mu_{\mathbb{CP}^n}$ should happen over some codimension two submanifold $S \subset \mathbb{CP}^n$. This would imply that the whole invariant measure of $SU(n+1)$ concentrates on a $U(n)$ fibration over $S$. This is the strategy of the proof that we will now make explicit. To this aim, we employ the explicit construction of the invariant measure over Lie groups given in [12]. In particular, the analysis of the geometry underlying the construction of the invariant measure for $SU(n)$ has been

performed in [17]. Fix a generalized Gell–Mann basis $\{\lambda_I\}_{I=1}^{n^2+2n}$ for the Lie algebra of $SU(n+1)$ as in [17]. Thus, the first $n^2$ matrices generate the maximal subgroup $U(n)$, the last one being the $U(1)$ factor, and, in particular, the matrices $\{\lambda_{(a+1)^2-1}\}_{a=1}^n$ generate the Cartan torus $T^n$. Then, the parametrization of $SU(n+1)$ can be obtained inductively as

$$SU(n+1) \ni g = h \cdot u, \tag{68}$$

where $u \in U(n)$ is a parametrization of the maximal subgroup, and

$$h = e^{i\theta_1\lambda_3} e^{i\phi_1\lambda_2} \prod_{a=2}^n [e^{i(\theta_a/\epsilon_a)\lambda_{a^2-1}} e^{i\phi_a\lambda_{a^2+1}}], \qquad \epsilon_a = \sqrt{\frac{2}{a(a-1)}}, \tag{69}$$

parametrizes the quotient. From $h$, one can construct a vielbein for the quotient as follows. Let $J_h$ be the Maurer–Cartan one-form of $SU(n+1)$ restricted to $h$. Then, set

$$e^l = \frac{1}{2}\mathrm{Tr}[j_h \cdot \lambda_{n^2+l-1}], \quad l = 1, \ldots, 2n. \tag{70}$$

They form a vielbein for $SU(n+1)/U(n) \simeq \mathbb{CP}^n$ so that

$$s_{\mathbb{CP}^n}^2 = \delta_{lm} e^l \otimes e^m, \tag{71}$$

$$d\mu_{\mathbb{CP}^n} = \det \underline{e} \tag{72}$$

are the metric and invariant measure, respectively, induced on $\mathbb{CP}^n$. In particular, one gets

$$\det \underline{e} = 2 d\theta_n d\phi_n \cos\phi_n \sin^{2n-1}\phi_n \prod_{a=1}^{n-1} [\sin\phi_a \cos^{2a-1}\phi_a d\theta_a d\phi_a]. \tag{73}$$

One can also write down the metric. Indeed, it has been shown in [17] that it is exactly the Fubini–Study metric for $\mathbb{CP}^n$ written in unusual coordinates. Since this is relevant for our analysis, let us summarize it. Let $(\zeta_0 : \cdots : \zeta_n)$ be the homogeneous coordinates and

$$\mathcal{K} = \frac{1}{2}\log(|\zeta_0|^2 + \ldots + |\zeta_n|^2) \tag{74}$$

be the Kähler potential. Fix a coordinate patch, say $U_0 = \{\underline{\zeta} : \zeta_0 \neq 0\}$ with the relative non-homogeneous coordinates $z_i = \zeta_i/\zeta_0$, $i = 1, \ldots, n$. When $\underline{z}$ varies in $\mathbb{C}^n$, the coordinate patch covers the whole $\mathbb{CP}^n$ with the exception of a real codimension two submanifolds defined by the hyperplane

$$S_n \equiv \mathbb{CP}^{n-1} = \{0 : \zeta_1 : \cdots : \zeta_n\}, \tag{75}$$

the so-called hyperplane at infinity. In these local coordinates, the Fubini–Study metric has components $g_{i\bar{j}} = \partial^2 \mathcal{K}/\partial z_i \partial \bar{z}_j$:

$$ds_{F-S}^2 = \frac{\sum_i dz_i d\bar{z}_i}{1 + \sum_j |z_j|^2} - \frac{\sum_{i,j} \bar{z}_i dz_i z_j d\bar{z}_j}{(1 + \sum_j |z_j|^2)^2}. \tag{76}$$

Following [17], let us introduce the change of coordinates

$$z_i = \tan\xi R_i(\underline{\omega}) e^{i\psi_i} \tag{77}$$

where $R_j(\underline{\omega})$ is an arbitrary coordinatization of the unit sphere $S^{n-1}$, $\psi_i \in [0, 2\pi)$, $\xi \in [0, \pi/2)$. In these coordinates,

$$ds^2_{F-S} = d\xi^2 + \sin^2 \xi \left[ \sum_i dR_i dR_i + \sum_i R_i^2 d\psi_i d\psi_i \right] - \sin^4 \xi \left[ \sum_i R_i^2 d\psi_i \right]^2. \tag{78}$$

In [17], it has been proved that this metric coincides with (71) after a simple change of variables, which, in particular, includes $\xi = \phi_n$. On the other hand, from (73), using

$$\int_0^{\pi/2 - \varepsilon} \cos \phi_n \sin^{2n-1} \phi_n d\phi_n = \frac{\cos^{2n} \varepsilon}{2n}, \tag{79}$$

we see that the measure over $\mathbb{CP}^n$ concentrates around $\phi_n = \xi = \pi/2$. Finally, since

$$\begin{aligned}
(1 : \tan \xi R_1(\underline{\omega}) e^{i\psi_1} : \cdots : \tan \xi R_n(\underline{\omega}) e^{i\psi_n}) \quad &= (1/\tan \xi : R_1(\underline{\omega}) e^{i\psi_1} : \cdots : R_n(\underline{\omega}) e^{i\psi_n}) \\
&\mapsto (0 : R_1(\underline{\omega}) e^{i\psi_1} : \cdots : R_n(\underline{\omega}) e^{i\psi_n})
\end{aligned} \tag{80}$$

when $\xi \to \pi/2$, we see that the concentration is on the hyperplane $S_n$ at infinity. Thus, if

$$\iota : S_n \hookrightarrow \mathbb{CP}^n \tag{81}$$

is the embedding of the hyperplane and if we look at $SU(n+1)$ as a fibration over $\mathbb{CP}^n$, we get that the whole measure concentrates on

$$\Sigma = \iota^*(SU(n+1)), \tag{82}$$

which is what we had to prove. □

**Remark 1.** *It is worth remarking that we are not saying the sequence of manifolds we have selected completely describes the concentration. Indeed, it is obvious that the concentration can take place on proper subspaces of the sequence. For example, (79) shows that the volume of the region $B_\varepsilon$ defined by $|\xi - \pi/2| > \varepsilon$, so in the complement of the concentration locus, it has volume vanishing as*

$$\sim e^{-n\varepsilon^2}. \tag{83}$$

*Let us now take $\varepsilon \to \varepsilon/\sqrt{N}$ and consider $N$ regions $B_{\varepsilon/N}^k$, $k = 1, \ldots, N$ associated to $N$ $\mathbb{CP}_n$ planes intersecting transversally in a point $p$ of our concentration locus. Then, $\bigcup_{k=1}^N B_{\varepsilon/N}^k$ has a volume of order $V_n \sim N e^{-n\varepsilon^2/N}$. Its complement is a subset of codimension $N$ of the concentration locus. In order to have $V_n \to 0$, it is sufficient that, for example,*

$$N e^{-n\varepsilon^2/N} \sim \frac{N}{n} \tag{84}$$

*to that we can consider $N \equiv N_n$ as dependent on $n$, with the condition that*

$$N_n \frac{\log n}{n} \to 0 \tag{85}$$

*when $n \to \infty$ with $\varepsilon$ fixed. Therefore, the codimension in general can diverge, and we get concentration loci of divergent codimension. This shows that it is not clear at all if a notion of optimal concentration can be defined.*

**Remark 2.** *In order to get uniform concentration in the sense of Gromov and Milman, we have to add a further hypothesis to our proposition, as already suggested by Formula (63). From Corollary 1, we see that if one normalizes the size of $SU(n+1)$ so that its coroots have length 2 (the standard*

*choice), then its scalar curvature is $r_n = \frac{n+3}{4}$. However, we can, in general, relax this condition and leave the length $|\check{\alpha}|$ of the coroots free. In this case, the scalar curvature becomes*

$$r_n = \frac{n+2}{|\check{\alpha}_n|^2}. \tag{86}$$

*Following [6], we see that we have a Levy family if $|\check{\alpha}_n|$ grows less than $\sqrt{n}$. It is interesting to notice that if we approximate the shape of the group as the product of n spheres of radius $|\check{\alpha}_n|$, then its diameter scales as $|\check{\alpha}_n|\sqrt{n}$. Thus, the uniform concentration is guaranteed if the diameter of the group grows less than $\sim n$. Since the dimension of $SU(n+1)$ is $d_n = n^2 + 2n$, we see that the condition is such that the diameter must grow less than $\sqrt{d_n}$, which is very similar to the case of the spheres.*

*Finally, this can also be understood from (79). Indeed, keeping the diameter fixed, we see that the $\varepsilon$ dependence is dominated by the therm $\cos^{2n} \varepsilon = (1 - \sin^2 \varepsilon)^n$. In place of rescaling the diameter, assume we rescale $\varepsilon$ in a n-dependent way, so $\varepsilon \to \varepsilon_n$, and assume that $\varepsilon_n \to 0$ when $n \to \infty$. Therefore, for large n, we have*

$$\cos^{2n} \varepsilon \sim e^{-n\varepsilon_n^2}, \tag{87}$$

*which converges to zero only if $n\varepsilon_n^2 \to \infty$. This means that $\varepsilon_n$ must decrease to 0 slower than $n^{-\frac{1}{2}}$, which is to say that $\varepsilon/|\check{\alpha}|$ must go to zero slower than $n^{-\frac{1}{2}}$ independently from how we allow $\varepsilon$ and $|\check{\alpha}|$ to vary separately with n.*

**Remark 3.** *The concentration metric in the form (78) becomes degenerate at the concentration locus when $\xi = \frac{\pi}{2}$, since one has to further fix one of the phases $\psi_j$. Nevertheless, if we consider a region $V_r$ of $\xi$-radius $\pi/2 - \xi = r$ around that locus, since the total measure is normalized to 1, we see from (79) that its volume is*

$$\mu(V_r) = 1 - \cos^{2n} r. \tag{88}$$

*As in [8], Section 2.1, we can use the inequality $\cos r \le e^{-\frac{r^2}{2}}$ for $0 \le r \le \frac{\pi}{2}$, so that*

$$\mu(V_r) \ge 1 - e^{-nr^2}, \tag{89}$$

*which gives us an estimation of how much the measure concentrates around the singular locus: for any fixed $r > 0$, the measure of $V_r$ converges exponentially to the full measure when n increases.*

It is worth mentioning that the limit topology depends not only on the topology of each space of the chain but also from the embeddings defining the sequence of groups. For example, we can replace the canonical embedding $U(n) \subset U(n+1)$ with the embeddings

$$U(n) \overset{J}{\hookrightarrow} SU(n+1) \subset U(n+1) \tag{90}$$

with

$$J(X) = \begin{pmatrix} X & \vec{0} \\ \vec{0}^t & \det X^{-1} \end{pmatrix}.$$

These embeddings lead to the result $SU(\infty)_J = U(\infty)_J$ for any limit topology we choose. Observe that if we use canonical embeddings, it is unknown whether the inductive limit $SU(\infty)$ is extremely amenable or not [21].

### 4.1.2. Odd Special Orthogonal Groups

The second classical series of simple groups is given by the odd-dimensional special orthogonal groups $SO(2n+1)$ of dimension $n(2n+1)$ and rank $n$. The center of the universal covering $Spin(2n+1)$ is $\mathbb{Z}_2$. The Lie algebra is the compact form of $B_n$, $n \ge 2$.

The invariant degrees are $d_i = 2i$, $i = 1, \ldots, n$ and the dimensions of the spheres generating the cohomology are $D_i = 4i - 1$. If we choose the standard normalization, a fundamental system of simple roots in $\mathbb{R}^n \simeq H_{\mathbb{R}}^*$ is given by $\alpha_i = \boldsymbol{e}_i - \boldsymbol{e}_{i+1}$, $i = 1, \ldots, n-1$, and $\alpha_n = \boldsymbol{e}_n$. The corresponding coroots are $\check{\alpha}_i = \alpha_i$ for $i = 1, \ldots, n-1$, and $\check{\alpha}_n = 2\alpha_n$. There are $p = n^2$ positive coroots, $n$ of which have length 2 and the others have a square length of 2. The volume of the torus is

$$V(T^n) = |(\boldsymbol{e}_1 - \boldsymbol{e}_2) \wedge (\boldsymbol{e}_{n-1} - \boldsymbol{e}_n) \wedge 2\boldsymbol{e}_n| = 2. \tag{91}$$

The Macdonald's formula thus gives

$$V(Spin(2n+1)) = \frac{2^{n(n+2)+1} \pi^{n(n+1)}}{\prod_{i=1}^n (2i-1)!}, \tag{92}$$

so that

$$\frac{V(Spin(2n+1))}{V(Spin(2n-1))} = \frac{2^{2n+1} \pi^{2n}}{(2n-1)!} \sim \sqrt{\frac{4\pi}{n - \frac{1}{2}}} \left( \frac{2\pi e}{2n-1} \right)^{2n-1}. \tag{93}$$

Since $\dim Spin(2n+1) - \dim Spin(2n-1) = 4n - 1$, we have

$$\left( \frac{V(Spin(2n+1))}{V(Spin(2n-1))} \right)^{\frac{1}{4n-1}} \sim \left( \frac{2\pi e}{2n} \right)^{1/2}, \tag{94}$$

which shows the same behavior as for the unitary groups. Again, in order to understand how concentration works, we have to do some geometry.

**Proposition 2.** *Consider the sequence of simple groups $Spin(2n+1)$ endowed with the bi-invariant metric. Set $B_n = S^{2n} \times S^{2n-1} \equiv Spin(2n+1)/Spin(2n-1)$ so that $Spin(2n+1)$ looks as a $Spin(2n-1)$-fibration over $B_n$. Finally, let $S_n$ be a bi-equator of $B_n$ (the Cartesian product of the equators of the two spheres), and*

$$\iota : S_n \hookrightarrow B_n \tag{95}$$

*the corresponding embedding. Then, in the limit $n \to \infty$, the invariant measure $\mu_n$ of $Spin(2n+1)$ concentrates on the codimension two subvariety*

$$\Sigma_n = \iota^*(Spin(2n+1)), \tag{96}$$

*in the sense of Definition 2.*

**Proof.** Since the proof is much simpler than in the previous case, we just sketch it, leaving the details to the reader. By using the methods in [12], in a similar way as before, it is easy to prove that the invariant measure $d\mu_n$ factorizes as

$$d\mu_{Spin(2n+1)} = d\mu_{Spin(2n-1)} \times dm_{S^{2n}} \times dm_{S^{2n-1}}, \tag{97}$$

where $dm$ is the Lebesgue measure (an independent way to see it is to notice that the measure is invariant under translations, and the quotient of the fibration is completed exactly through the action of translations under a subgroup). Therefore, since it is well known that the Lebesgue measures on the spheres concentrate over the equators, again, the measure $d\mu_n$ concentrates on a $Spin(2n-1)$ fibration over a codimension two submanifold of $S^{2n} \times S^{2n-1}$. $\square$

### 4.1.3. Symplectic Groups

The compact form $USp(2n)$ of the symplectic group of rank $n$ has dimension $2n^2 + 2$. Its center is $\mathbb{Z}_2$ and its Lie algebra is the compact form of $C_n$, $n \geq 2$. The invariant degrees are

the same as for $SO(2n+1)$, so they have the same sphere decomposition. In the standard normalization, the roots of $USp(2n)$ are the coroots of $SO(2n+1)$ and vice versa. Therefore, we have $n^2 - n$ coroots of length $\sqrt{2}$ and $n$ of length 1. The volume of the torus is

$$V(T^n) = |(\boldsymbol{e}_1 - \boldsymbol{e}_2) \wedge (\boldsymbol{e}_{n-1} - \boldsymbol{e}_n) \wedge \boldsymbol{e}_n| = 1, \tag{98}$$

and the volume of the group is

$$V(Usp(2n)) = \frac{2^{n^2} \pi^{n(n+1)}}{\prod_{i=1}^{n}(2i-1)!}. \tag{99}$$

Again, we get

$$\left( \frac{V(USp(2n))}{V(USp(2n-2))} \right)^{\frac{1}{4n-1}} \sim \left( \frac{2\pi e}{2n} \right)^{1/2}. \tag{100}$$

**Proposition 3.** *Consider the sequence of symplectic groups $USp(2n)$ endowed with the bi-invariant metric. Set $S^{4n-1} \equiv USp(2n)/USp(2n-2)$ so that $USp(2n)$ looks as an $USp(2n-2)$-fibration over $B_n = S^{4n-1}$. Finally, let $S_n$ be an equator of $B_n$, and*

$$\iota : S_n \hookrightarrow B_n \tag{101}$$

*be the corresponding embedding. Then, in the limit $n \to \infty$, the invariant measure $\mu_n$ of $Spin(2n+1)$ concentrates on the codimension one subvariety*

$$\Sigma_n = \iota^*(USp(2n)), \tag{102}$$

*in the sense of Definition 2.*

The proof is the same as for the spin groups.

### 4.1.4. Even Special Orthogonal Groups

The last series is given by the even-dimensional special orthogonal groups $SO(2n)$ of dimension $n(2n-1)$ and rank $n$. The center of the universal covering $Spin(2n)$ is $\mathbb{Z}_2 \times \mathbb{Z}_2$ if $n = 2k$ and $\mathbb{Z}_4$ if $n = 2k+1$. The Lie algebra is the compact form of $D_n$, $n \geq 4$. The invariant degrees are $d_i = 2i$, $i = 1, \ldots, n-1$, $d_n = n$ and the dimensions of the spheres generating the cohomology are $D_i = 4i-1$, $i = 1, \ldots, n-1$, $D_n = 2n-1$. If we choose the standard normalization, a fundamental system of simple roots in $\mathbb{R}^n \simeq H_{\mathbb{R}}^*$ is given by $\alpha_i = \boldsymbol{e}_i - \boldsymbol{e}_{i+1}$, $i = 1, \ldots, n-1$, and $\alpha_n = \boldsymbol{e}_{n_1} + \boldsymbol{e}_n$. The corresponding coroots are $\check{\alpha}_i = \alpha_i$ for $i = 1, \ldots, n$, and all have length $\sqrt{2}$. There are $p = n^2 - n$ positive coroots. The volume of the torus is

$$V(T^n) = |(\boldsymbol{e}_1 - \boldsymbol{e}_2) \wedge (\boldsymbol{e}_{n-1} - \boldsymbol{e}_n) \wedge (\boldsymbol{e}_{n-1} + \boldsymbol{e}_n)| = 2. \tag{103}$$

Thus,

$$V(Spin(2n)) = \frac{2^{n^2+1} \pi^{n^2}}{(n-1)! \prod_{i=1}^{n-1}(2i-1)!}, \tag{104}$$

and

$$\frac{V(Spin(2n))}{V(Spin(2n-2))} = \frac{2(2\pi)^{2n-1}}{(2n-2)!} \sim \sqrt{\frac{4\pi}{n-1}} \left( \frac{2\pi e}{2n-2} \right)^{2n-2}. \tag{105}$$

Since $\dim Spin(2n) - \dim Spin(2n-2) = 4n-3$, we have

$$\left( \frac{V(Spin(2n))}{V(Spin(2n-2))} \right)^{\frac{1}{4n-3}} \sim \left( \frac{2\pi e}{2n} \right)^{1/2}, \tag{106}$$

which, again, shows concentration.

**Proposition 4.** *Consider the sequence of simple groups $Spin(2n+2)$ endowed with the bi-invariant metric. Set $B_n = S^{2n+1} \times S^{2n} \equiv Spin(2n+2)/Spin(2n)$ so that $Spin(2n+2)$ looks as a $Spin(2n)$-fibration over $B_n$. Finally, let $S_n$ be a bi-equator of $B_n$, and*

$$\iota : S_n \hookrightarrow \mathbb{B}_n \tag{107}$$

*be the corresponding embedding. Then, in the limit $n \to \infty$, the invariant measure $\mu_n$ of $Spin(2n+2)$ concentrates on the codimension two subvariety*

$$\Sigma_n = \iota^*(Spin(2n+2)) \tag{108}$$

*in the sense of Definition 2.*

This exhausts the classical series. Further considerations can be made by using the Riemannian structure analyzed in Section 3. Here, we limit ourselves to notice that in principle, we can construct a huge number of Levy families as a consequence of the Theorem in Section 1.2, page 844 of [6]:

**Corollary 2.** *Let $Y_i = (X_i, g_i, \mu_i)$ be a family of compact Riemannian spaces with natural normalized Riemannian measures. Assume there is a positive constant $c > 0$ such that definitely $r_i \geq c$, where*

$$r_i = \inf \rho^{(i)}(\tau, \tau) \tag{109}$$

*with the information taken in the set of all tangent vectors of unit length. Consider any sequence of positive constants $c_i$ such that*

$$\lim_{i \to \infty} c_i = \infty. \tag{110}$$

*Then, the new family*

$$\tilde{Y}_i = (X_i, \tilde{g}_i, \mu_i), \qquad \tilde{g}_i = \frac{1}{c_i} g_i \tag{111}$$

*is Levy.*

**Proof.** Obviously, $\tilde{r}_i = c_i r_i$. Since definitely $r_i \geq c$, we have $\lim_{i \to \infty} \tilde{r}_i = +\infty$. □

## 5. Further Comments and Conclusions

In a companion paper, [1], we have introduced the notion of "concentration locus" for sequences of groups $G_n$, $G_n \subseteq G_{n+1}$, endowed with normalized invariant measures. Then, we have shown in which sense the mapping of the concentration locus on a set, through its action on that set, governs the concentration of the measure on the set and eventually determines the presence of a fixed point. Here, we have seen how a concentration locus can be determined for the classical series of compact Lie groups. This loci can have an unboundedly increasing codimension and determine probes for analyzing the action of some infinite dimensional Lie groups on (not necessarily) compact sets or manifolds. We remark that the question about the extreme amenability of $SU(\infty)$, for example, is still an open problem [21]. The result we obtained for the classical series can be easily extended to more general sequences of compact Lie groups.

**Proposition 5.** *Let $\{G_n\}$ be a family of connected compact Lie groups of the form*

$$G_n = G_n^{(1)} \times \cdots \times G_n^{(k_n)} \times T^{s_n}/Z_{G_n}, \tag{112}$$

*where $T^{s_n}$ is a torus of dimension $s_n$, $G_n^{(1)} \times \cdots \times G_n^{(k_n)}$ is the product of $k_n$ compact connected simple Lie group, and $Z_{G_n}$ is a finite subgroup. Suppose that among the factors of $G_n^{(1)} \times \cdots \times G_n^{(k_n)}$, there exists a finite dimensional connected compact Lie group $G_0$ common to all n. Alternatively, assume that $s_n \neq 0$ for $n > n_0$. Then, there exists at least a finite dimensional compact manifold*

*K admitting an equicontinuous action of $G_\infty$, which is taken with the inductive limit topology without fixed points.*

*If at least one of the $G_n^{j_n}$ determines a classical sequence $\{G_n^{j_n}\}_{n \in N}$ of compact Lie groups, then a concentration locus of $G_n$ is obtained restricting the factors $G_n^{j_n}$ to the corresponding concentration loci.*

The proof is simple and is sketched in Appendix A. In the first part, obviously, $K = G_0$ or $K = S^1$. It generalizes the known result that $U(\infty)$ is not extremely amenable. The second part is just a corollary of our results in the previous sections.

It would be interesting to relate the concentration of the measure around the concentration loci to the phenomenon of optimal transport. We expect such a connection to be governed by the way the process of concentration around concentration loci is realized in our examples. We plan to investigate such connection in a future work.

**Author Contributions:** Conceptualization, P.U. and S.C.; methodology, P.U. and S.C.; investigation, P.U. and S.C.; writing—original draft preparation, P.U.; writing—review and editing, S.C. All authors have read and agreed to the published version of the manuscript.

**Funding:** This research received no external funding.

**Data Availability Statement:** Not applicable.

**Acknowledgments:** The second author gratefully acknowledges partial support from the projects MEGABIT—Università degli Studi di Catania, PIAno di inCEntivi per la RIcerca di Ateneo 2020/2022 (PIACERI), Linea di intervento 2.

**Conflicts of Interest:** The authors declare no conflict of interest.

## Appendix A. Proof of Proposition 5

In this section, we provide a sketch of the proof of Proposition 5. Let us first assume that among the factors of $G_n^{(1)} \times \cdots \times G_n^{(k_n)}$, there exists a finite dimensional connected compact Lie group $G_0$ common to all $n$, say $G_n^{(j_n)}$ for some given sequence $j_n$, $n = 1, 2, \ldots$. The fact that these are common factors just means that there does exist a compact simple Lie group $G$ and a sequence of group isomorphisms

$$A_n : G_n^{(j_n)} \longrightarrow G. \tag{A1}$$

These isomorphisms determine actions of $G_n$ on $G$ all equivalent to the natural action of $G$ on itself. This shows that $K = G$ is a compact manifold on which all $G_n$ act without fixed points, and then, their inductive limits do the same.

In a similar way, if $s_n > 0$ for $n > n_0$, we can select in each $T^{s_n}$ at least a circle isomorphic to $S^1$. In this way, each $G_n$ acts as $U(1)$ on this $S^1$, and we can identify $K = S^1$ as the compact group on which the inductive limit of the family $G_n$ acts without fixed points.

Finally, assume that at least one of the $G_n^{j_n}$ determines a classical sequence $\{G_n^{j_n}\}_{n \in N}$ of compact Lie groups. Therefore, Propositions 1–4 ensure that the invariant measures on $\{G_n^{j_n}\}_{n \in N}$ concentrate on a suitable non-trivial invariant locus $\Sigma_{j_n} \subset G_n^{j_n}$. Since the measure on $G_n$ factorizes as the direct product of the measures on the factors, this implies that the invariant measure on $G_n$ concentrates at least on the product of $\Sigma_{j_n}$ times the remaining factors, which are then quotiented with the discrete subgroups. This completes the proof of the proposition.

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
