# Peer review of "Macdonald Formula, Ricci Curvature, and Concentration Locus for Classical Compact Lie Groups"

_axioms, doi:10.3390/axioms11060245_

Round 1

Reviewer 1 Report

The paper is interesting . The presented results are modern and important. It is calculated explicity  a concentration  locus for some classical topological groups .  The Ricci curvature of such gropu\ps is also  computed.

All calculations are correct. Some new ideas are presented. The paper is well organized and well written.

Author Response

We thank all referees for their comments.

List of changes: Since for some reasons the package “trackchanges” didn’t work correctly, in revising the manuscript we reported all changes in color blue (with the exception of the corrections of typos), to help the referees and editors to identify them easily. In any case we also report the changes here below.
We have revised and expanded both the abstract and the introduction, where we also included new references.
We have corrected two typos in Definition 1. Just before Theorem 1 we have also corrected π1(G) to π1(GΓ).
In section two, after formula (2.1), in formula (2.2) and in formula (2.3) we changed notations. Before formula (2.5) we added a new reference. After formula (2.8) and in formulas (2.9), (2.10), (2.11) and (2.14), we have changed notations. After (2.14) and in (2.15) we have renamed the irreducible representation (irrep). In formulas (17), (18) and in the proof of Corollary 1 we changed notations. Before and after (2.22) we renamed the irrep. Just above (2.23) we added two references. Just above (2.30) we renamed the irrep and added a reference. Above (2.33) we renamed the irrep, while below we added a reference.
In section 3, below (3.6), we corrected SU (n) to SU (n + 1). Above (3.26) we corrected the specification of the region Bε. In remark 2, above and in formula (3.29) we corrected the notation. The same has been done just above formula (3.52), in formula (3.52), and in the proof of Corollary 2. In the proof of Proposition 2 we added a comment that makes it independent on the results of previous work.
In section 4, just below Proposition 5, we referred to Appendix A.
We added Appendix A, with the proof of Proposition 5.
We have extended the bibliography, changed letters to numbers and reordered the bibliography in alphabetic order.

Finally, for the present article we suggest the following MCS codes
MSC: 28A75, 60A10, 22E15, 22F10, 54H20
and key words
Key Words: Lie groups; Invariant measures; Concentration; Topological Dynamics

We hope that with the above modifications, including a revision of the English, our manuscript can
be accepted for publication on Axioms.

Reviewer 2 Report

The authors analyze the concentration locus, as defined in the first section, relative to the Macdonald formula and Ricci Curvature. The methods are generally well described and the derivations adequate. I support the publication of the manuscript after some minor corrections.

  1. The introduction requires more context e.g. the first paragraph mentions "pioneering work"  but no description. Missing dot at the end of paragraph.
  2. The introduction requires a bit more citations e.g. Ricci curvature and lie groups. e.g. DOI: 10.1007/BF02418270 and ISBN 978-3-540-15279-8. Also probably relating to curvature and its applications in physics could enhance the importance of the publications e.g. https://doi.org/10.1103/PhysRevB.98.155419
  3. after eq 2.8 maybe it is prudent to change the letter used for the vielbein from j to something else,   e.g. t for tetrad or l?  
  4. Similaly Ric_{ij} is a bit confusing as i and c look like indices and variable respectively? maybe change to some other version of R?

Author Response

We thank all referees for their comments. Please see the attachment.

Reviewer 3 Report

Nevertheless, some disadvantages are still not be resolved:

(1) The abstract needs to be revised and expanded in the revision.

(2) The introduction needs to be revised and expanded in the revision.

(3) In the introduction, the authors did not provide a strong motivation for the paper and the obtained results. In addition, they should discuss the main contributions of their work in detail after the motivation part. Then, they should summarize the main structure of their paper in brief at the end of the introduction.

(4) The English writing of the paper is required to be improved. Please check the manuscript carefully for typos and grammatical errors. The reviewer found some typos and grammatical errors within this manuscript, which have been excluded from my review. In addition, the English structure of the article, including punctuation, semicolon, and other structures, must be carefully reviewed.

Author Response

(The authors gave the same response as above.)

Reviewer 4 Report

In the manuscript entitled "MACDONALD FORMULA, RICCI CURVATURE, AND CONCENTRATION LOCUS FOR CLASSICAL COMPACT LIE GROUPS" the authors attempt to provide an analysis on the "concentration locus" using the well known MacDonald's formula and the Ricci curvature. There are a number of comments both from a technical perspective as well as other general points that I believe must be considered:

1- The affiliation of the authors should be given in the first page and not after the bibliography  

2- The abstract is unusually short and general; it does not contain the main message for the study. It just simply repeats the title of the manuscript.

3- The referral to the references throughout the manuscript are not given in the standard format. In a scientific manuscript, the references have to be numbered and the use of letters is not a proper way of citation.

4- The content of the manuscript is mathematically accurate, however the main message of the manuscript cannot be deducted until the conclusion section. Though the research is scientifically sound, in multiple parts of the manuscript, including in the conclusion section, the authors refer to their previous work.

5- The generalization of the main idea of this work to wider aspects compact Lie groups is missing in the manuscript.; A point in which can greatly improve the work.  

6- In parts of the manuscript, the mathematical proof of some statements are absent. For example in proposition number 5, the authors leave the proof for the reader. I suggest the authors add an appendix to the manuscript in which they can express the results of earlier work as well as add mathematical derivations to the statements mentioned in the main text.

In conclusion the paper must be improved in order to be published in Axioms.  

Author Response

(The authors gave the same response as above.)

Round 2

Reviewer 3 Report

The authors have to reply to their comments in a polite way.

Reviewer 4 Report

The authors have clearly addressed most of my comments and the manuscript can be published in its present form.